# Modelling COVID-19 transmission in a hemodialysis centre using simulation generated contacts matrices

Mohammadali Tofighi[1]*, Ali Asgary[1], Asad A. Merchant[2], Mohammad Ali Shafiee[2], Mahdi M. Najafabadi[1], Nazanin Nadri[1], Mehdi Aarabi[2], Jane Heffernan[3], Jianhong Wu[4]

**1** ADERSIM (Advanced Disaster, Emergency, and Rapid Response Simulation), York University, Toronto, Ontario, Canada, **2** University Health Network (UHN), Toronto, Ontario, Canada, **3** Modelling Infection and Immunity Lab, York University, Toronto, Ontario, Canada, **4** LIAM (Laboratory for Industrial and Applied Mathematics), York University, Toronto, Ontario, Canada

* tofighim@yorku.ca

**Data Availability Statement:** All relevant data are within the paper.

## Abstract

The COVID-19 pandemic has been particularly threatening to patients with end-stage kidney disease (ESKD) on intermittent hemodialysis and their care providers. Hemodialysis patients who receive life-sustaining medical therapy in healthcare settings, face unique challenges as they need to be at a dialysis unit three or more times a week, where they are confined to specific settings and tended to by dialysis nurses and staff with physical interaction and in close proximity. Despite the importance and critical situation of the dialysis units, modelling studies of the SARS-CoV-2 spread in these settings are very limited. In this paper, we have used a combination of discrete event and agent-based simulation models, to study the operations of a typical large dialysis unit and generate contact matrices to examine outbreak scenarios. We present the details of the contact matrix generation process and demonstrate how the simulation calculates a micro-scale contact matrix comprising the number and duration of contacts at a micro-scale time step. We have used the contacts matrix in an agent-based model to predict disease transmission under different scenarios. The results show that micro-simulation can be used to estimate contact matrices, which can be used effectively for disease modelling in dialysis and similar settings.

## Introduction

The COVID-19 pandemic has had a massive impact on all facets of public health. As a disease, it is particularly threatening to patients with end-stage kidney disease (ESKD) on intermittent hemodialysis and their care providers. Full compliance with mitigation strategies for reducing infection spread, such as physical distancing and avoidance of high-density areas including healthcare facilities is usually not feasible. Patients with ESKD who receive in-centre hemodialysis, face unique challenges. They are obligated to be at a dialysis unit three or more times a week, where they are confined to specific seating and tended to by dialysis nurses in close proximity. Dialysis patients are already at higher risk of experiencing more severe manifestations of disease due to their poor functional immune status and their increased burden of co-

**Funding:** The University Health Network; Public Health Agency of Canada; Canadian Institute of Health Research, Ontario; Research Funds, National Science and Engineering Research Council of Canada. The funders had no role in study design, data collection and analysis, decision to publish, or preparation of the manuscript.

**Competing interests:** The authors have declared that no competing interests exist.

morbidities [1]. Furthermore, ESKD patients who have not been tested positive for the SARS-CoV-2 are not able to effectively quarantine and must travel to the hospital for their regular treatments, often using public transportation [2], hence increasing the risk of viral spread to the population at large, and more specifically to the dialysis unit staff. Not surprisingly, the first COVID-19 associated death in the United States was a hemodialysis patient at the Northwest Kidney Center in the State of Washington [3].

Many in-centre dialysis facilities have experienced outbreaks among dialysis patients, physicians, and nursing staff leading to nursing and physician shortages in an already overburdened healthcare system [4]. These outbreaks have downstream effects throughout the healthcare centres in which they are based. Therefore, several proposals have been put forward to reduce the risk of infectious spread in dialysis units during the COVID-19 pandemic.

It is important to understand the potential impact of an outbreak and the pathways through which the virus can propagate in the dialysis unit environment. Currently, only a limited number of modelling studies of the SARS-CoV-2 spread and containment in hospital settings, particularly hemodialysis units are available. This study aims to fill this gap by using a combination of discrete event and agent-based simulation to model the operations of a large Canadian dialysis setting in Toronto and the impacts of the infection and infection protection measures.

## Background

The maintenance hemodialysis (MHD) population is highly susceptible to infection by SARS-CoV-2 and at the same time, there is a high risk of outbreaks of COVID-19 in MHD centres [5]. MHD patients are at increased risk of COVID-19 and its complications because they tend to be older and have multiple comorbidities including hypertension and suppressed immune systems [5]. Furthermore, chronic kidney disease (CKD) can be considered as a cardiovascular risk equivalent, and COVID-19 that enhances the risk of cardiovascular events may synergize with pre-existing cardiovascular risk factors in CKD patients [6,7]. Hemodialysis patients attend dialysis facilities several times a week each time spending between 3 to 5 hours for their treatment [8]. Home-based dialysis has a clear advantage during a pandemic as patients do not need to travel and maintain physical distancing. The need for frequent trips to dialysis units and unavoidable patient clustering during dialysis shifts, further increase the risk of viral transmission [9]. Transportation to dialysis centres may require interactions with transportation personnel and other passengers in the public transit systems. In addition, MHD unit daily operations incorporate several patient-to-patient and patient-to-caregiver operations that increase the risk of COVID-19 transmission. This is because of synchronous schedules of dialysis that have patients entering and exiting the department at the same time, as well as close contacts with healthcare workers that have a similar type of interaction with other patients. Moreover, because of the unique characteristics of MHD patients and MHD units, it is more difficult to prevent and control infectious diseases in these settings compared to imposing control measures for the general population and in other settings [10].

Early reports from Canadian healthcare facilities also confirm that MHD patients are highly susceptible to COVID-19 infections and vulnerable to their severe consequences. According to the Canadian Institute for Health Information [11], about 620 MHD COVID-19 positive patients (excluding Quebec) were hospitalized in Canada from January to March 2020 (Table 1). Currently, all MHD with suspected or confirmed COVID-19 infections are being triaged to hospitals [12].

As such, the COVID-19 pandemic presents a particular challenge to MHD patients and in-centre units in hospitals as the risk of transmission to the medical staff, facility workers,

**Table 1. COVID-19 hospitalizations of dialysis patients (including intensive care unit admissions) and associated characteristics in Ontario and Canada (January to March 2020).**

| Description | Canada (Excluding Quebec) | Ontario |
|---|---|---|
| Hospitalizations Number with COVID-19 diagnosis (confirmed and suspected) | 620 | 408 |
| Hospitalizations Number with confirmed COVID-19 diagnosis | 453 | 289 |
| Hospitalizations Percentage of COVID-19 hospitalizations with a confirmed diagnosis | 73.1 | 70.8 |
| Percentage of male | 57.4 | 56.1 |
| The average total length of stay (in days) | 4.1 | 3.9 |
| Median Age (in years) | 62 | 62 |
| Disposition Percentage discharged home | 69.7 | 69.9 |
| Disposition Percentage transferred to other inpatient care | 12.4 | 11 |
| Disposition Number died in the facility | 89 | 65 |
| Hospitalizations Number with COVID-19 diagnosis (confirmed and suspected) | 69.7 | 69.9 |

Source of data: Canadian Institute for Health Information [11].

patients, and family members is considerably higher [13]. While previous studies have examined the impacts of disasters such as earthquakes, floods, and hurricanes on dialysis units and their service clients and patients [14], studies on the impacts of a global pandemic are very limited. In this section, we will review the status quo, current challenges and concerns, and emerging solutions and recommendations for the SARD-Cov-2 outbreak in MHD units.

## COVID-19 and hemodialysis patients and dialysis units

Near 14.5 million people are living with ESKD globally [15]. Most of these individuals (more than 95%) dialyze in specialized facilities [8]. Compared to the general public, dialysis patients are at short-term mortality of 20% or higher with COVID-19 [16]. According to the Canadian Institute for Health Information [17], 40,289 Canadians (excluding Quebec) were living with ESKD at the end of 2018. In Ontario alone, there are about 13,330 chronic dialysis patients [18]. In Canada, MHD is provided by not-for-profit Renal Programs in large academic in-centre hemodialysis associated with medical schools and community hospitals [19]. Ontario has 27 Renal Programs treating MHD patients and delivers HD at approximately 100 sites [18]. Many personnel in hemodialysis facilities are also involved in providing dialysis services that should not be halted even during a pandemic.

Infection rates in dialysis centres are twice those of in-home dialysis patients [16]. Only a small portion of the MHD patients (ranging between 1 to 5%) fall under the category of home-based dialysis [20]. The expansion of home-based dialysis is now a public health discussion. Since implementing some of the basic control measures may not be feasible in dialysis units and considering the lower risk of COVID-19 exposure in home-based dialysis, initiatives such as the ESKD Prospective Payment System and the Executive Order on Advancing American Kidney Health are underway in some countries to reform the existing dialysis services from centre-based to home-based services to reduce the risks.

In earlier phases of the COVID-19 pandemic, not all hemodialysis facilities had not adapted their workflow processes to the situation. Some dialysis facilities did not have testing capabilities and thus, before the infected people were identified, several others (patients and staff) were already infected [21]. Limited studies in different parts of the world such as Wuhan, China [8],

United Kingdom and Ontario, Canada, and France [22] show the infectious rate of COVID-19 amongst MHD had been somewhere from 1% to 16.1%. Mortality rates in MHD patients with COVID may be as high as 20% [16]. Additionally, a significant number of dialysis patients live in long-term care facilities [23]. Liu, Ghai [21] examined the prevalence of COVID-19 infection at two dialysis centres and found that patients who lived in long-term care facilities suffered from much higher infection rates of 37% and 88%.

### COVID-19 challenges for dialysis centres

An outbreak in hemodialysis centres can cause staff shortages and thus, puts a strain on health professionals and technicians that run these facilities [24]. This can lead to increased waiting time for dialysis and more exposure, or otherwise, patients who skip their dialysis shifts due to fear from the pandemic, putting them at risk of fluid overload and metabolic emergencies [25]. One study in a hospital in Madrid found that 20% of the healthcare staff in the Nephrology department were diagnosed with COVID-19 [1]. Resource scarcity is compounded by a general shortage of medical equipment, and PPEs during a pandemic [8]. Therefore, the protection of staff working in dialysis units is very important.

The ability of a dialysis centre to provide effective physical distancing for all its members is constrained by several factors including the physical layout of the unit which cannot be reconfigured without a major overhaul [21]. Space is limited, and there is little flexibility in the scheduling to allow the following special precautions for cohorting COVID-19 infected patients. Most dialysis units operate at maximum capacity. The nature of care for these patients requires close proximity to nurses operating with the HD machine. These nurses may interact with 3 or 4 patients simultaneously. Although contacts with healthcare providers can be limited, they cannot be eliminated. Asymptomatic spread in the HD unit is another challenge [1]. As a consequence of a relatively suppressed immune system, the proportion of asymptomatic infection in MHD populations may be higher than average [26,27].

According to Hsu and Weiner [16] to mitigate the impact of COVID-19 in dialysis units, we should focus on three main areas including dialysis facilities, transportation, and patients' communities. Although MHD centres have little control over the latter two, they can mitigate transmission risks within their facilities. In order to address these challenges, hospitals, nephrology departments, and hemodialysis centres have adopted recommended protocols and guidelines internally and externally. These guidelines focus on three transmission routes: transmission from patients; transmission from healthcare professionals, and transmission from the surface [1].

## Materials and methods

### Index hemodialysis centre

We build our simulation platform and models based on the University Health Network (UHN) in-centre dialysis program. UHN has one of the largest in-centre hemodialysis programs in Toronto, situated at Toronto General Hospital (TGH). This unit serves 308 patients, of which 278 do intermittent hemodialysis three to six days a week, and 30 are nocturnal/evening dialysis patients who dialyze three days a week. There are two separate but contiguous units at TGH ("Hemo-east" and "Hemo-west", separated by a corridor) with a total of 55 dialysis chairs that serve 55 patients at a time (Fig 1). The westward treatment area is 580 m$^2$ consisting of a 165 m$^2$ corridor and 245 m$^2$ for dialysis stations with an average area of 9 m$^2$ per station. The eastward treatment area is 1235 m$^2$ and consists of a 299 m$^2$ corridor and 301 m$^2$ for dialysis stations with an average area of 9.7 m$^2$ per station. Regular dialysis stations are

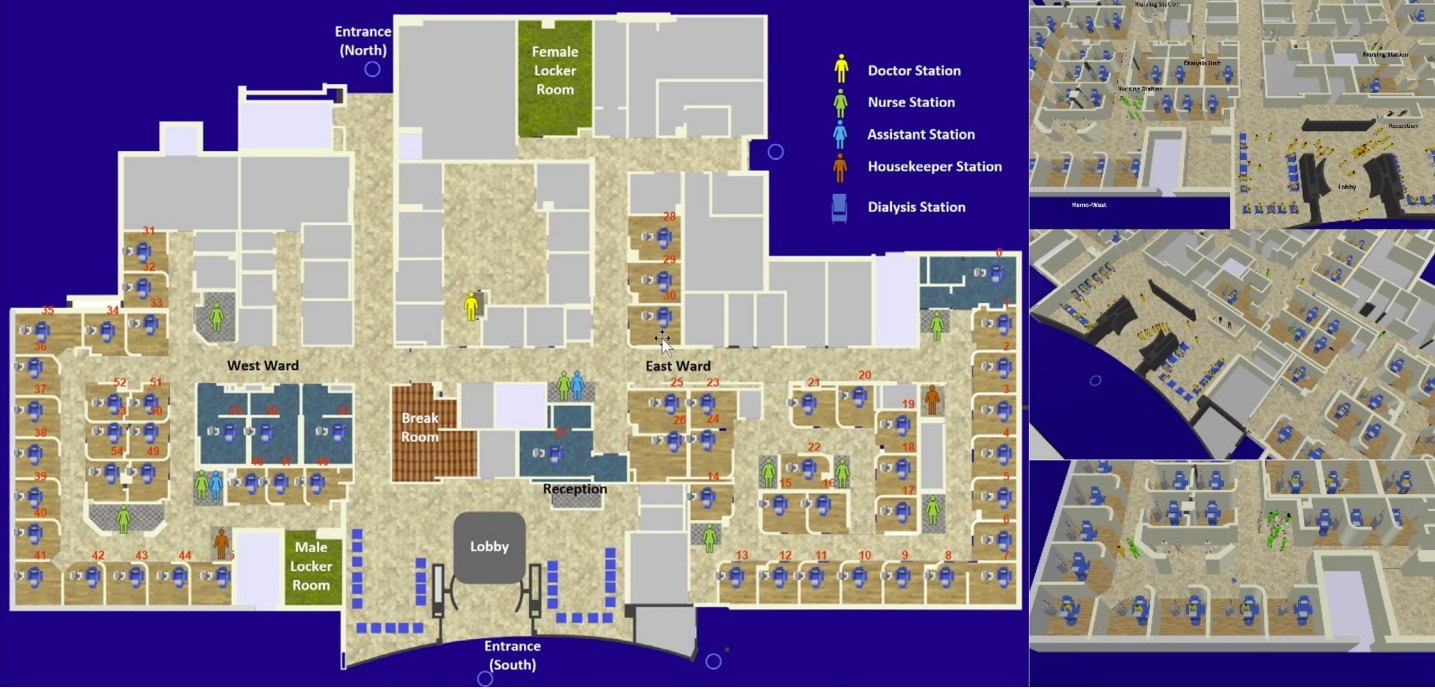

**Fig 1. 2D and 3D models of Toronto General Hospital, hemodialysis departments developed for the simulations.**

divided by curtains and isolate dialysis stations have divider partitions. Regular dialysis stations are located in the perimetral and middle of the wards.

The hemodialysis unit is staffed by nephrologists, nurses, assistants (technicians), pharmacists, dieticians, social workers, and other clerical and support staff. There are three daily shifts and a nocturnal shift from Mondays to Saturdays. The working time in the hemodialysis department had been scheduled in three 10-hour shifts that overlap each other about two and four hours in the nocturnal and daily shifts, respectively. Nurses are assigned to specific dialysis stations, and not to patients. Patients are randomly assigned to the stations and there is no consistency regarding which stations they are assigned. Patients will likely be assigned different nurses each day. Thus, the nurses will not consistently take care of the same patients on the next appearance of the patient in the centre. Patients are neither assigned to the same stations, although some have their preferred spot. It is done to maintain maximum flexibility and interchangeability. Nurses will often finish their shifts while their patients are being dialyzed. They will therefore handover patient care to the nurses from the next shift. Nurses rotate between patients during dialysis intermittently, assessing blood pressure (BP-automatically checked by the machine every 15 min to 30 min), blood flows, and circuit pressures. The nurses congregate at the nursing station closest to their patients. After the patient is served and left the station, the housekeeping staff cleanse the chair/bed, replace linens, and clean the area. The nurse is responsible for cleaning the patient chart and BP cuff.

Before, or between the shifts, hemodialysis assistants (HAs) set up the dialysis machines. This includes putting on dialyzers, putting tubes onto the machine, ensuring the machines are cleaned and sterilized according to the protocol, and ensuring other needed equipment is available at the station. There are three HAs per shift for each of the two wards (east ward and west ward). When they are not actively preparing machines, the HAs usually sit at nursing stations.

Ten different nephrologists are responsible for the 12 shifts (six on each of the two wards). A nurse practitioner (NP) assists the nephrologists in each shift. Nephrology fellows are also assigned to round in the unit on behalf of the most responsible physician (MRP). The MRP spends approximately 1–2 hours in the dialysis unit, once a week, while the NP and/or the nephrology fellow spend approximately 1–2 hours per shift rounding on the patients on both wards. When the NP or fellow is not actively seeing patients, they may stay in their office area or simply leave the unit.

Table 2 shows the current key infection mitigation measures used in the dialysis centre. Nurses, physicians, and other staff in the unit wear face masks and face shields, especially when interacting with patients at close range. All health care providers don gloves and gowns when interacting with patients who are on droplet precautions (any patients who screen positive or are positive for any respiratory infection). Patients are asked to socially distance (2 meters) when in the waiting room of the dialysis unit. All patients are screened as they enter, and those who screen positive are prioritized to enter the unit and are tested with nasopharyngeal swabs for viral detection.

In the hemodialysis units, there are five rooms with negative pressure isolation that are used for COVID-positive or presumptive positive patients. If there are more, eight other stations have been identified where COVID patients will be preferentially dialyzed. All stations have at the very least curtains to separate if needed.

## Dialysis centre simulation and contacts calculator model

In this study, we used a two-stage approach to model the dialysis centre operation and disease spread among the staff and patients. We used a combination of discrete event pedestrian and agent-based modelling to simulate various agents' movement in our study hemodialysis centre and an agent-based disease transmission model to simulate disease transmission based on the

**Table 2. Key infection mitigation measures used in the dialysis centre.**

| Measures | Description |
|---|---|
| Screening | Fever screening with a thermometer and infrared camera, Inquiring about contacts history and related symptoms (cough, sputum, sore throat) by a standard checklist |
| Triage | Separate clinics for persons who have either fever or respiratory illness, and who have a history of travelling abroad or visiting high-risk areas, located in front of the entrance to the outpatient clinic and emergency room. |
| Access control | The study site minimized the number of entrances to Dialysis buildings and restrict visitors with patients |
| Universal mask-wearing | All healthcare workers, employees, patients, and visitors are obligated to wear masks in the hospital |
| Diagnostic | Real-time Reverse Transcription Polymerase chain reaction (RT-PCR) for any patients with related symptoms or suspected findings without specific causes |
| Isolation | Isolation room with negative pressure used for Patients with either fever or pneumonia Healthcare workers for those patients are required to wear appropriate personal protective equipment (PPE). |
| Space management | Rearrangement of beds, machines, furniture to enhance physical distancing |
| Signage and communication | Physical distancing signage on the floors |
| Shifts and staff workflow management | Changing patients shifts and staff to stay with their own cohorts |
| Staff gatherings and meetings | Cancellation of meetings and gatherings of staff for meetings and lunch breaks |
| Hygiene practices | provide alcohol dispensers in patients waiting rooms and advise patients to use them; MHD patients are recommended to wash their hands and fistula arm before starting dialysis and to thoroughly disinfect the puncture areas |

contact matrices generated by the pedestrian model. The contact matrices were created based on three different types of interactions: patient to patient, patient to staff (health care providers), and staff to staff.

To model the workflow and disease transmission in the dialysis department, we used discrete event and agent-based simulation capabilities of the AnyLogic® simulation modelling software (version 8.7.0). AnyLogic® provided an environment that enables simulation of the geometry of the centre and processes such as scheduling, workflow, peoples, and their attributes.

Contacts between the patients and the dialysis staff can be loci of infectious spread. The probability of virus transmission is a function of the number of contacts, average contact duration, and the distance of the subject with the substance [28]. To model disease spread, we recorded agent movements and created contact matrices that include this information. The contact matrix is one of the mathematical components used for disease modelling among different groups of individuals in an environment. This matrix meets the reciprocity of contacts which makes the contacts rate matrix symmetric [4]. To obtain such a matrix for our case study (the dialysis unit), we simulated the real setting of the dialysis centre and its workflow processes. Agent movements were simulated using the Pedestrian library in AnyLogic® software. Agents moved according to predefined physical rules for the simulation. They interacted with the surrounding objects, including other agents, walls, and escalators, and avoided potential collisions. The user of the model (hereafter: user) can assign individual properties, preferences, and states to these agents–here, pedestrian agents. Using the pedestrian model, we simulated the movements of staff and patients based on their schedules and workflows. Using this simulation, we found the number of contacts and the duration in which the agents were in proximity. These values are presented as contacts matrices, that can be applied to disease transmission models.

In general, we can define active or inactive agents in AnyLogic® agent-based simulation platform. Active agents (i.e., human agents) in our case are the staff and patients, and inactive agents are dialysis machines, beds/chairs, and furniture in the department (Fig 2). We defined MRPs, nephrologists, nurses, assistants, clerks, housekeepers, and patients as the active agents, and beds/chairs as inactive agents. Although staff such as pharmacists, dieticians, social workers, etc. work in the same unit, for convenience and simplicity, we assumed that they had very limited contacts with patients. The model boundary in this simulation (environment of agents) is the dialysis department comprising of common hallways and areas, and wards that were embedded in the model in full scale and detail. Active agents could interact and move in the predefined areas according to their assigned duties. The movement of the agents was controlled by the built-in Pedestrian library of the AnyLogic® software.

Workflows for staff and patients were assigned based on the treatment process (Fig 3). These workflows ran in parallel (i.e., simultaneously) through the simulation run using an event manager. Table 3 presents the number of staff and their shift assignments. We assumed that all staff and patient schedules are fixed on a weekly time basis. All in-shift staff enter through the north entrance, check their daily assignment at a board in the eastward corridor entrance, and go to their locker room to change into uniform and get ready. Female and male locker rooms are located in the east and west wards respectively. There is one station for all physicians in the east ward and several stations for other staff in both wards (Fig 1). Patients come from the main entrance (south of the building), register at the clerk's desk, and wait in the lobby area for their treatments' turn. The admission is on a first come first serve basis (FIFO model). The patients sit in the lobby based on the order they entered the environment. After a dialysis station is prepared by an assistant, the patient is directed to the ward by a receptionist clerk. The dialysis treatment takes about 3.5 to 4.5 hours. During the treatment period, the nurses supervise the process. The nurses circulate between the dialysis stations and their

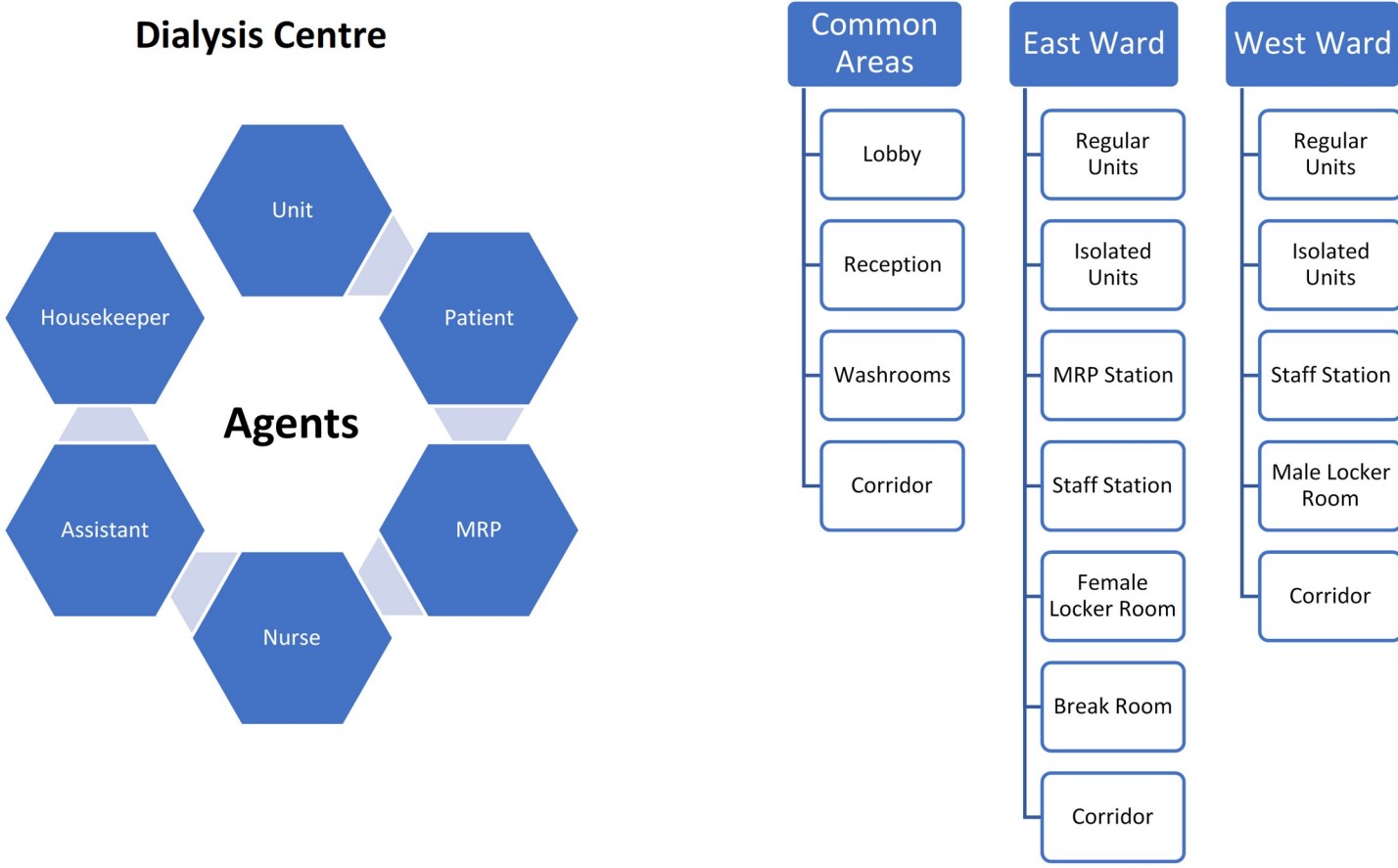

**Fig 2. General structure of the agent-based simulation model (Left: Agents, Right: Environment Detail).**

assigned patients. We assumed that checking the patients takes place once every 15-minutes. At the end of the dialysis, the nurse disconnects the dialysis machine from the patient and the patient returns to the lobby, to depart. Some patients stay in the lobby for a while before they leave the dialysis unit. Patients receive treatment 3 to 6 times a week, four shifts a day, and we assumed are familiar with the process.

In the agentSource block of each agent in the workflow (Fig 3), agents are created at the beginning of the simulation in their virtual homes, which are just outside the hospital environment. The agents of a certain shift start to attend the centre at the start of their shift. Staff find their shift assignment (number of dialysis stations that should be served during the shift) and go to their locker room based on their gender then going to the nearest station to their assignments. At this location, agents check to find if their assigned station(s) need their service or not. This check is regulated by a virtual event manager every one second. The chain of steps for a dialysis station is 1) cleaning by a housekeeper; 2) preparing the machine by an assistant; 3) getting a patient; 4) connecting the machine to the patient by a nurse; 5) multiple checkups of the patient by the assigned nurse during the service; and 6) disconnecting the machine from the patient by the nurse after the treatment is over. Physicians (MRPs) randomly visit patients that need advice. Patients come to the lobby, lineup, get screened at the reception and wait for a message from an assistant declaring that the dialysis station is ready. When patients reach their assigned station, they stay there until the end of the treatment. Once the treatment is over, the patients return to the lobby and wait for a short time and then and leave the

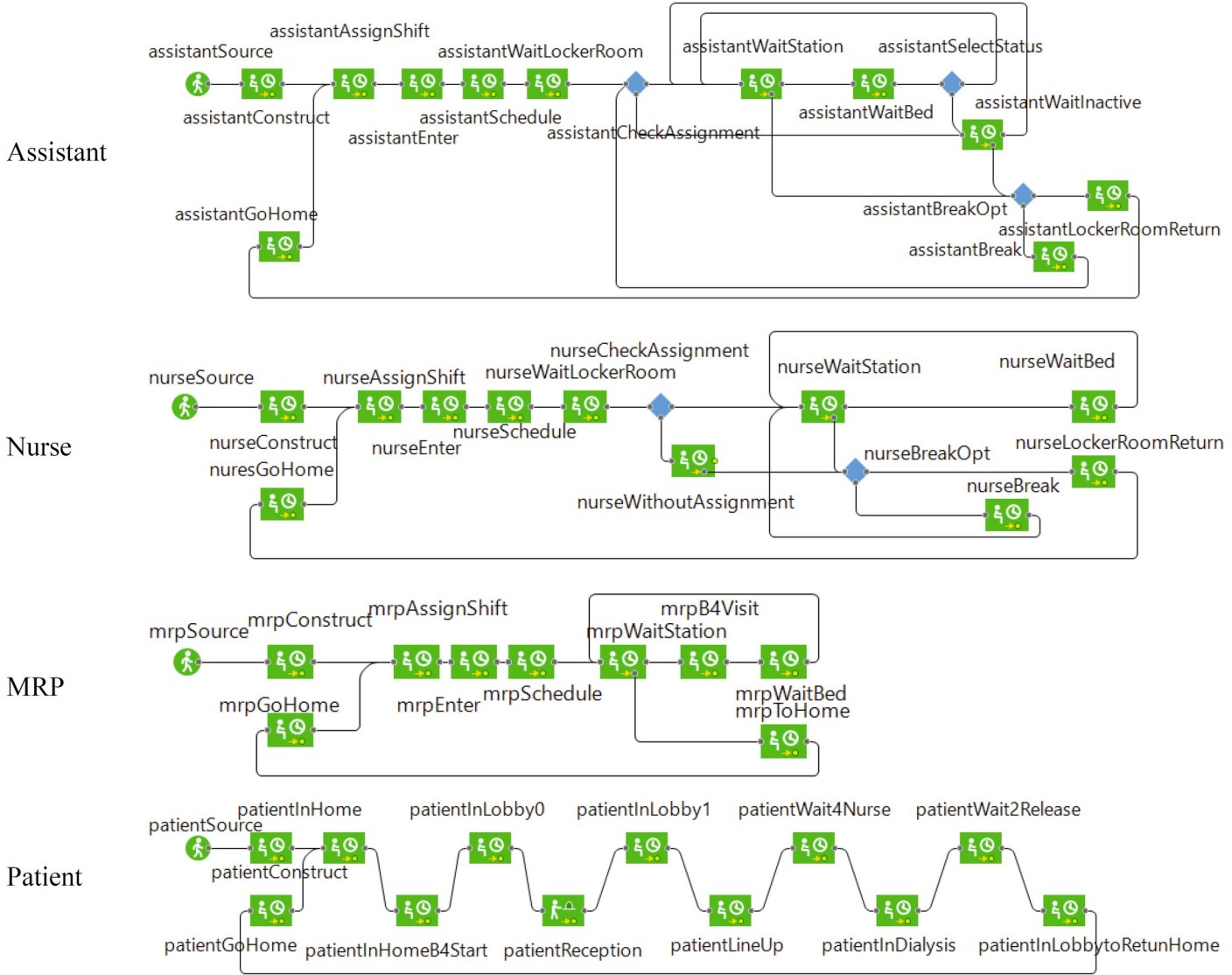

**Fig 3. Workflow of selected active agents in AnyLogic® pedestrian model.**

environment. There is a break time for the staff during their shift, during which the staff who wait in the centre with no assignment leave the area and go to the breakroom for a maximum of 40 minutes. These processes repeat during a workday to serve all incoming patients. Table 4 shows the used parameters in the model.

**Table 3. Weekly Schedule of healthcare providers and patients.**

| Staff | Total Number | Weekly Schedule | Number per shift per ward | Shift Assignment |
|---|---|---|---|---|
| MRP | 10 | 1 to 3 days | 1 or 2 | Random |
| Nephrologist/NP | 12 | 1 to 3 days | 1 or 2 | Random |
| Nurse | 110 | Everyday | 10 ~ 25 | Random |
| Assistant | 30 | Everyday | 3 ~ 8 | Random |
| Housekeeper | 10 | Everyday | 2 | Fixed |
| Clerk | 6 | Everyday | 2 for both | Fixed |
| Patient | 308 | 3 ~ 6 days | 55 for both | Random |

**Table 4. Parameters used in the agent-based simulation.**

| Agent | Parameter* | Value | Unit |
|---|---|---|---|
| Staff and Patients | Speed | 0.16 | m/s |
| Staff | Time in the Locker Room | 5–10 | minute |
| Staff | Break Time | 20–40 | minute |
| Assistant | Preparing Material Time in the Station | 2–4 | minute |
| Assistant | Staying Time in the Dialysis Station | 1–2 | minute |
| Nurse | Visiting Time in the Dialysis Station | 3–5 | minute |
| Nurse | Time of Connecting and Disconnecting Patient and the Machine | 20–40 | minute |
| Physician | Visiting Time in the Dialysis Station | 3–5 | minute |
| Patient | Staying Time in the Screening Process | 0.5–1.5 | minute |
| Patient | Staying Time in the Dialysis Station for Treatment | 210–270 | minute |
| Patient | Staying Time in the Lobby before Leaving the Hospital | 4–6 | minute |

*. Parameter values are based on the existing workflow of the unit.

By using the Pedestrian library for the active agents in this model, the movement of the patients and staff and their proximity is simulated during each shift in the dialysis centre. We directly calculated close contacts between all people in the centre as well as the time-length of the contacts. In the dialysis model, in total, 486 active agents and 58 inactive agents were defined. Due to the weekly cyclic and random schedule of the active agents, every agent had the probability of contacts with the 485 active agents in the same or different categories. For example, a nurse could come to close contacts with 109 nurses, 30 assistants, 308 patients, 22 physicians, 10 housekeepers, 6 clerks, and all beds/chairs in the dialysis department during a week. To calculate the contacts, we assumed proximity less than 2 meters is a contact. For every agent, we defined an array for each category of agent instances with the size of the total members of that agent to store the history of the contacts between the agents. A simple sample of this array for an agent (we call it centre agent) between eight agents from three categories has been presented in Fig 4.

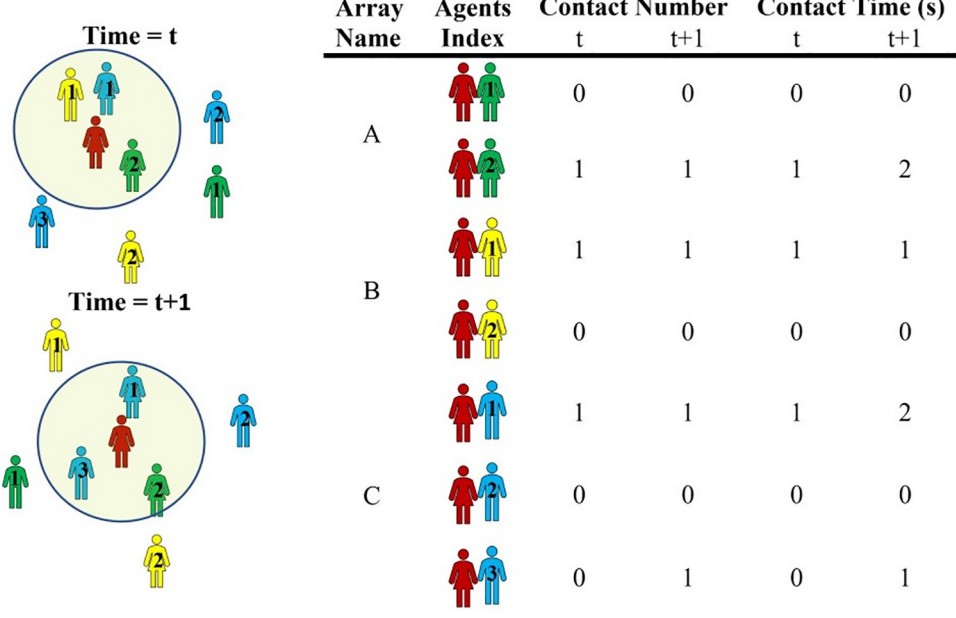

**Fig 4. Contact array at two successive times.**

Three arrays (A, B, and C) have been defined for these categories. Every element of each array represents one of the eight agents that the centre agent may become to close contacts with. This element comprises a three-element array that stores the time of connection, number of contacts with that agent, and the cumulative time of the connection. In every second for the centre agent, the nearest (< 2 meters distance) agents are geometrically defined. These agents have been located inside the circle with the centre point of the central agent in Fig 4. For each agent in proximity, the close contacts time is stored in the corresponding element of the array of the centre agent. If this agent at the previous time step was not in proximity, one close contact is added to the corresponding counter of the agent otherwise this contact is not encountered as a new contact. One second is also added to the corresponding element of the cumulative time of connection. Simulating for an arbitrary time, we can record the number and duration of contact for all agents in the arrays. Using such arrays for all agents of a category, the statistical variables of contact parameters between different or the same category of agents can be evaluated and used for the disease transmission model.

### Disease transmission model

Although the contacts calculator model can precisely simulate the process and provide micro-scale contacts parameters in our case study dialysis unit, it takes a long time to be run and generate the results for long-term simulation. To simulate the disease transmission quicker and in more flexibility for examining various scenarios, we developed another agent-based model. We used a modified version of the SEIR (Susceptible, Exposed, Infected, and Recovered) disease transmission model in this study. Other than the usual compartments of susceptible and exposed, infectious, and recovered, we have added additional compartments to include asymptomatic, pre-symptomatic, and symptomatic cases as well as self-isolation, hospitalization, and deceased compartments.

Fig 5 shows our agent-based model' state chart for each person including staff and patients. In this model, all persons are initially susceptible. Once an individual is infected and arrives in the hospital, it starts contaminating others based on the estimated contact rates that s/he has with other individuals and the probability of infections. These values vary depending on the type of person. If positive disease status is known, the person will be isolated, and the isolated individuals will no longer able to transmit the virus. The agent-based parameters used to run the base model are listed in Table 5.

## Results

Depending on the prespecified input parameters, the model generates a multitude of different results. In this section, we present results from a few simulation experiments and scenarios. We first present the result of the dialysis centre workflow that generated the contact matrices for us and then use the generated matrices in a sample experiment of our disease transmission model.

### Contacts model and simulation results

To obtain the parameters of the contacts generator model, we ran the dialysis unit simulation model in one-second time steps for a full week, using two scenarios: a) "base model"; and b) a model without a break schedule (following COVID-19 guidelines that strongly recommends not to have any unnecessary gatherings of the healthcare workers and the staff). In the base model (i.e., normal schedule scenario), all workflow steps in Fig 3 are done, while in the second scenario, break and gathering in the break room are omitted. Repeating the scenario runs or running the model for a long time did not change the parameters of the contact significantly.

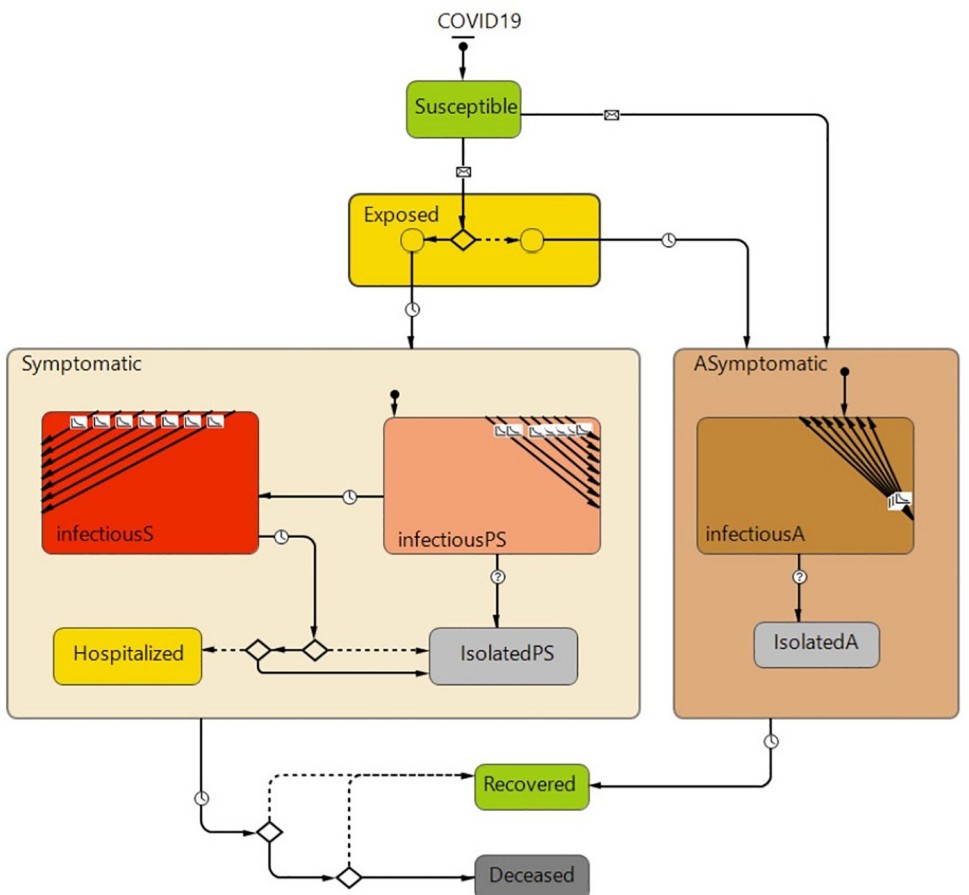

**Fig 5. Disease transmission model of the dialysis unit.**

Fig 6 illustrates the duration of close contacts between agents in the same or different categories in terms of cumulative hourly (sum of 3600 seconds) and daily (sum of 24 hours) time series.

As can be seen in Fig 6, between the time 13:00 and 18:00, as well as between 21:00 and 23:00 every day, when the schedules of staff from different shifts overlap, the time and number of contacts between agents increases. At other times, the hourly parameters of contacts remain slightly constant. At the nocturnal shift, the contacts are reduced due to the limited capacity of overnight patient admission to one unit of the department. At this time, the number of patients, nurses, and assistants is almost cut in half, but it is assumed that the number of clerks remains the same. The hourly length of contacts between clerks and the reduced number of contacts between other categories are well captured by the model. Comparing the assistant-to-assistant contacts and assistant-to-clerk contacts (Fig 6B) highlights the effect of break time in the connection parameters. If no gathering happens during the break times (13:00 to 14:00, 16:00 to 17:00, and 00:00 and 1:00), the number of contacts between different categories of staff is reduced. However, if agents do not leave their stations during the break time, the overall contacts between the agents of the same categories or the same station are increased in this time window.

Fig 7 presents the time series of the hourly and daily cumulative time and numbers of connections during a one-week simulation. Daily charting starts from Monday at 7:00 AM, and

**Table 5. Parameters of the agent-based disease transmission model.**

| Name | Value | Source |
|---|---|---|
| Transmission probability (infected symptomatic) | 0.14 | [29] |
| Recovery period (symptomatic) (day) | 12 | [30] |
| Asymptomatic incubation period (day) | 5.47 | [30,31] |
| Transmission probability PS (pre-symptomatic) | 0.05 | [30] |
| Transmission probability A (asymptomatic) | 0.14 | [29] |
| Recovery period A (asymptomatic) (day) | 9 | [32] |
| Patients hospitalization rate | 0.61 | [33,34] |
| Patients death rate | 0.5 | [31,34] |
| Pre symptomatic period (day) | 2.63 | [35] |
| Pre symptomatic rate | 0.5 | [31,34] |
| Pre symptomatic incubation period (day) | 2.4 | [30] |
| Infected random patient | 1 | Based on scenario |
| Infected random nurse | 0 | Based on scenario |
| Infected random MRP | 0 | Based on scenario |
| Infected random assistant | 0 | Based on scenario |
| Infected random clerk | 0 | Based on scenario |
| Infected random nephrologist | 0 | Based on scenario |
| Infected random housekeeper | 0 | Based on scenario |
| Number of replications | 500 | - |

day 7 of the model simulation represents Sunday when the unit is closed. As expected, the hourly contact parameters have fluctuations during the day and days of a week due to the randomness of both the staff work and patient treatment schedules. However, the daily averages of the contact parameters have fewer fluctuations. We can calculate matrices of the average time of contact and the number of contacts between all agent categories in any desired time span. We call this matrix micro-scale contacts matrix (MSCM) which can be calculated in every arbitrary time span. Assuming the daily values of contact parameters affect disease transmission, we use the daily contact matrix in our transmission model. Table 6 summarizes the contact parameter in the normal schedule and without break condition. These contact matrices are symmetric, as expected [4]. As explained above in the methods section, the contact parameters of the same category staff are generally increased when the break is omitted from the workflow.

## Disease transmission model and simulation results

Based on the modified SEIR disease transmission model that we developed for the agents and based on the contacts matrices developed by the simulation model, we ran this simulation for a base scenario (a normal condition with minimum protection and COVID-19 infection). We then focused on the impact of random testing of staff and patients on disease propagation. We used our model to examine the impact of break time on the results. Fig 8 and Table 7 summarize these model results for 45 days of simulation. At the start of these simulations, a random asymptomatic person from each category comes to the hospital and the spread of disease among all people is evaluated. Each simulation has run 500 times using different random seeds and the average results are presented here. For the testing scenario, five patients and five staff randomly tested for COVID-19 each week and if the test result is positive, they will be isolated until they are recovered. In Table 7 the maximum percentage of exposed individuals and the time that this maximum happens are presented.

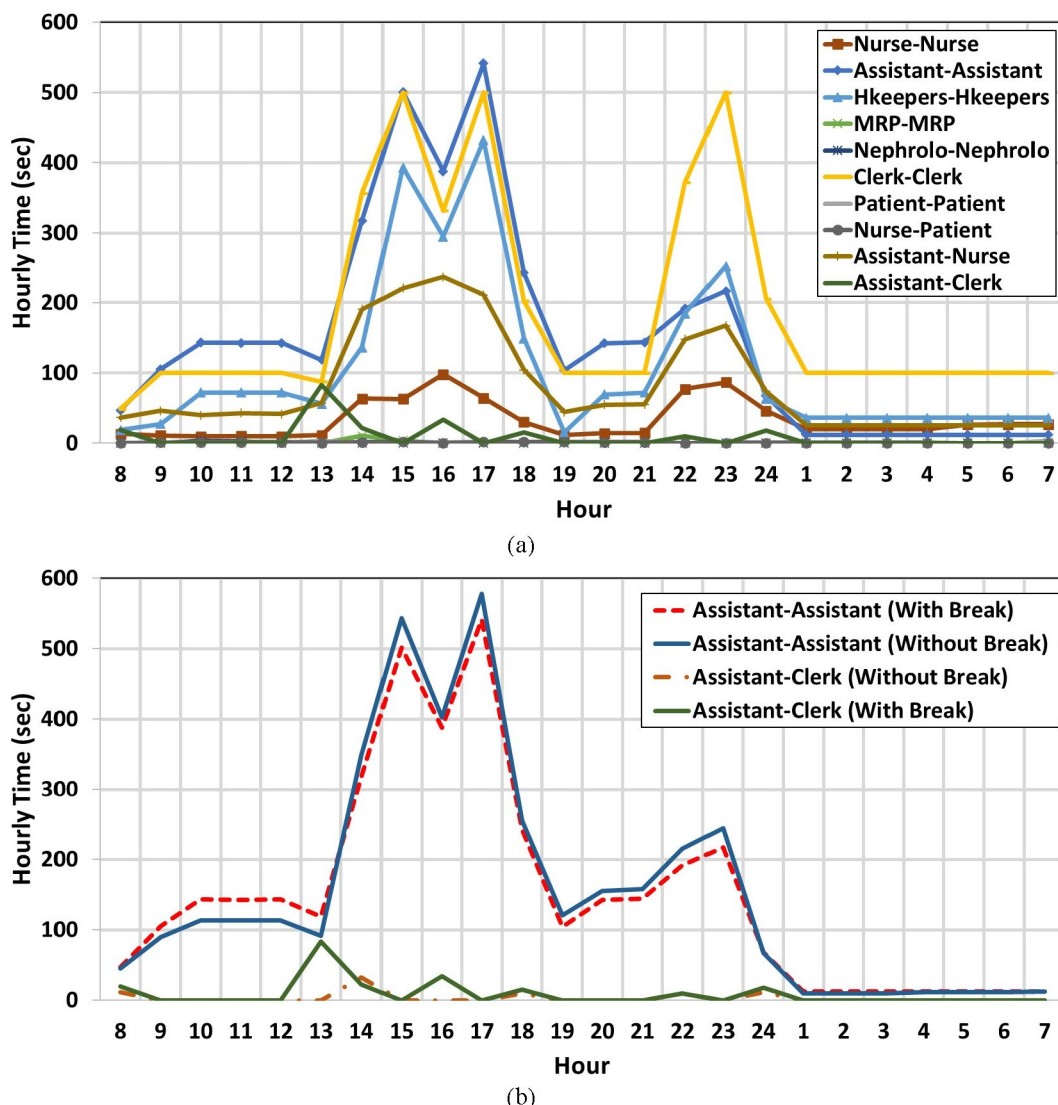

**Fig 6. Time series of hourly cumulative time of connections between agents.** a) normal schedule b) normal and without break time scenario.

MRPs and nephrologists have the least connection with the patients and staff with a shorter working time than other staff, and it is assumed they are not gathering in the break room with other staff. So, they have less probability of infection in the dialysis unit. They also work in different shifts and their connection together rarely happens. Therefore, all physicians are less likely to get infected, and if they are infected but are asymptomatic, their disease affects few individuals and remains limited to their assigned dialysis unit. The existence of an asymptomatic clerk, housekeeper, assistant, or nurse creates a condition that over 50 percent of staff in the unit would get infected. The time of reaching the maximum number of exposed people is shorter in this scenario.

An asymptomatic nurse or assistant brings the riskiest condition of disease spread in the dialysis unit. The time of reaching the maximum number of exposed people is shorter in this case. Comparing the columns *Pat.Tes.* and *Pat.* in Table 7 shows that random weekly testing of about 0.5% of all individuals can reduce all exposed individuals by 2%. Examining this scenario

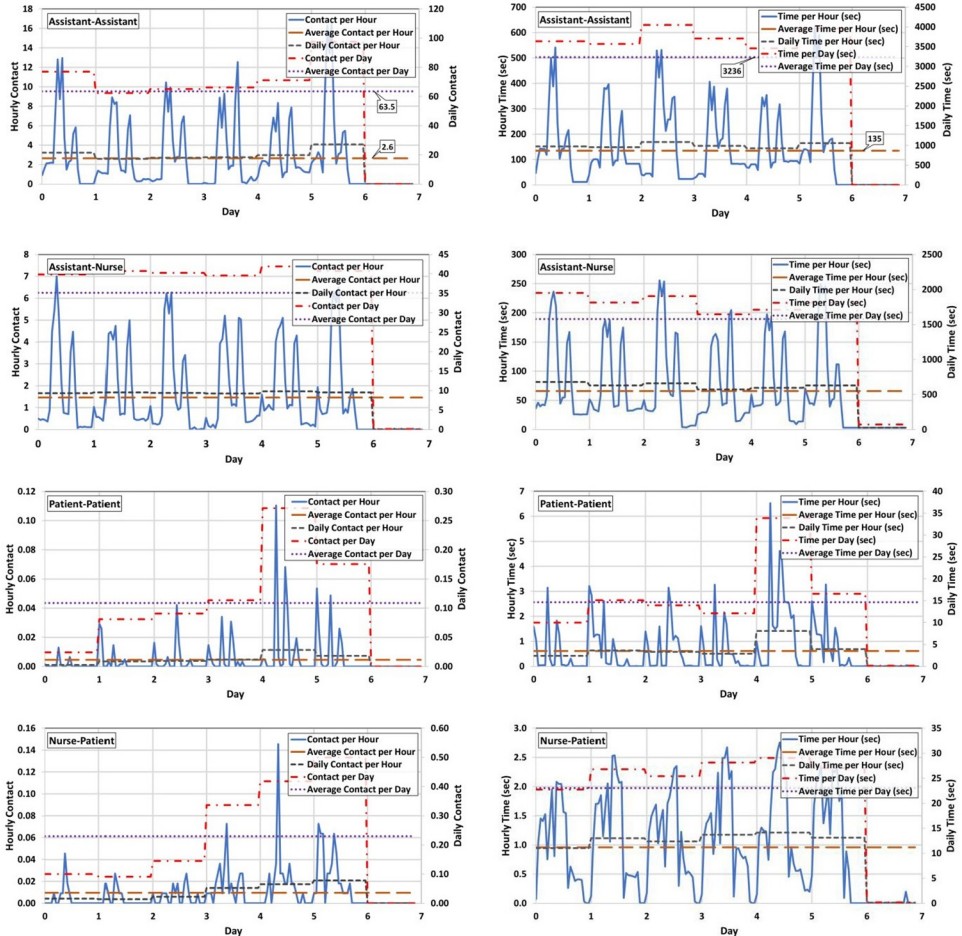

**Fig 7. Time series of the hourly/daily cumulative number (left charts) and time (right charts) of connections during a one-week simulation of the normal schedule of the dialysis department.**

with more random testing showed that random testing can reduce disease transmission that is in conformance with other studies [36]. It reduces the daily infection rates. Staff in dialysis units are also very more vulnerable. Our results show that asymptomatic staff poses more risk than asymptomatic patients. These results suggest that testing and disease transmission prevention measures for staff can be even more important and effective than for the patients. Results of the second scenario (no gathering during the breaks) are presented in Table 7. According to these results, if an asymptomatic nurse enters the dialysis unit (column *Nur. NoBr.*), the number of exposed individuals who are working with the nurse does not increase significantly. However, the results are different for clerks. In the no-break condition, clerks do not go to the break or lounge room, thus they do not experience close contact with other staff. In the case of an asymptomatic patient in the no-break condition, the number of exposed individuals does not significantly vary. This scenario confirms the importance of measures that reduce disease transmission between staff once more.

## Discussion

In this paper, we firstly provided some background related to the current situation, issues, and challenges of the SARS-CoV-2 infection in the Hemodialysis departments. Subsequently, the

**Table 6. Average micro-scale contact matrices (MSCM) during a one-week simulation.**

| | Patient | Clerk | Housekeeper | Assistant | Nurse | MRP | Nephrologist |
|---|---|---|---|---|---|---|---|
| *Average Time per Day (sec)—Base model* | | | | | | | |
| Patient | 17.0 | 3.5 | 15.2 | 3.5 | 26.8 | 0.4 | 0.0 |
| Clerk | 3.5 | 4871 | 121.33 | 169 | 188 | 0.0 | 0.0 |
| Housekeeper | 15.2 | 121 | 2876 | 284 | 177 | 0.0 | 0.0 |
| Assistant | 3.5 | 169 | 284 | 3775 | 1840 | 2.3 | 1.2 |
| Nurse | 26.8 | 188 | 177 | 1840 | 819 | 1.2 | 1.1 |
| MRP | 0.4 | 0.0 | 0.0 | 2.3 | 1.2 | 2.3 | 11.7 |
| Nephrologist | 0.0 | 0.0 | 0.0 | 1.2 | 1.1 | 11.7 | 4.6 |
| *Average Contact per Day* | | | | | | | |
| Patient | 0.12 | 0.03 | 0.70 | 0.12 | 0.23 | 0.00 | 0.00 |
| Clerk | 0.03 | 1.87 | 3.15 | 5.13 | 5.95 | 0.00 | 0.00 |
| Housekeeper | 0.70 | 3.15 | 2.3 | 20.5 | 7.7 | 0.00 | 0.00 |
| Assistant | 0.12 | 5.13 | 20.5 | 74.1 | 41.1 | 0.12 | 0.12 |
| Nurse | 0.23 | 5.95 | 7.7 | 41.1 | 16.6 | 0.05 | 0.02 |
| MRP | 0.00 | 0.00 | 0.00 | 0.12 | 0.05 | 0.00 | 0.35 |
| Nephrologist | 0.00 | 0.00 | 0.00 | 0.12 | 0.02 | 0.35 | 0.00 |
| *Average Time per Day (sec)—Without Break* | | | | | | | |
| Patient | 15.2 | 3.5 | 15.2 | 3.5 | 28 | 0.4 | 0.3 |
| Clerk | 3.5 | 5095 | 98 | 89 | 85 | 0.0 | 0.0 |
| Housekeeper | 15.2 | 98 | 2878 | 292 | 196 | 0.0 | 0.0 |
| Assistant | 3.5 | 89 | 292 | 3804 | 1878 | 2.3 | 1.2 |
| Nurse | 28.0 | 85 | 196 | 1878 | 799 | 1.2 | 1.1 |
| MRP | 0.4 | 0.0 | 0.0 | 2.3 | 1.2 | 2.3 | 11.7 |
| Nephrologist | 0.3 | 0.0 | 0.0 | 1.2 | 1.1 | 11.7 | 4.3 |
| *Average Contact per Day* | | | | | | | |
| Patient | 0.12 | 0.02 | 0.70 | 0.12 | 0.35 | 0.00 | 0.00 |
| Clerk | 0.02 | 0.70 | 1.75 | 2.22 | 2.10 | 0.00 | 0.00 |
| Housekeeper | 0.70 | 1.75 | 2.8 | 13.8 | 7.6 | 0.00 | 0.00 |
| Assistant | 0.12 | 2.22 | 13.8 | 77.1 | 41.1 | 0.12 | 0.12 |
| Nurse | 0.35 | 7.6 | 7.6 | 41.1 | 15.4 | 0.05 | 0.05 |
| MRP | 0.00 | 0.00 | 0.00 | 0.12 | 0.05 | 0.00 | 0.35 |
| Nephrologist | 0.00 | 0.00 | 0.00 | 0.12 | 0.05 | 0.35 | 0.00 |

results of the two developed models of a typical dialysis department were presented. These models were simulating the department's operations and disease transmission. The first model provided the real operation of the dialysis department, considering the schedules of work and treatment, the entrance of patients and staff, their stay in the lobby, stations, locker rooms, units, and movement and order of workflow in the dialysis department. The workflow of the patients, physicians, nurses, assistants, clerks, and housekeepers was simulated as a parallel and in-order process using the pedestrian library of the AnyLogic® software. The real number and time of contact for the agents in the hospital were evaluated and presented as a contacts matrix that was used for the simulation of disease transmission.

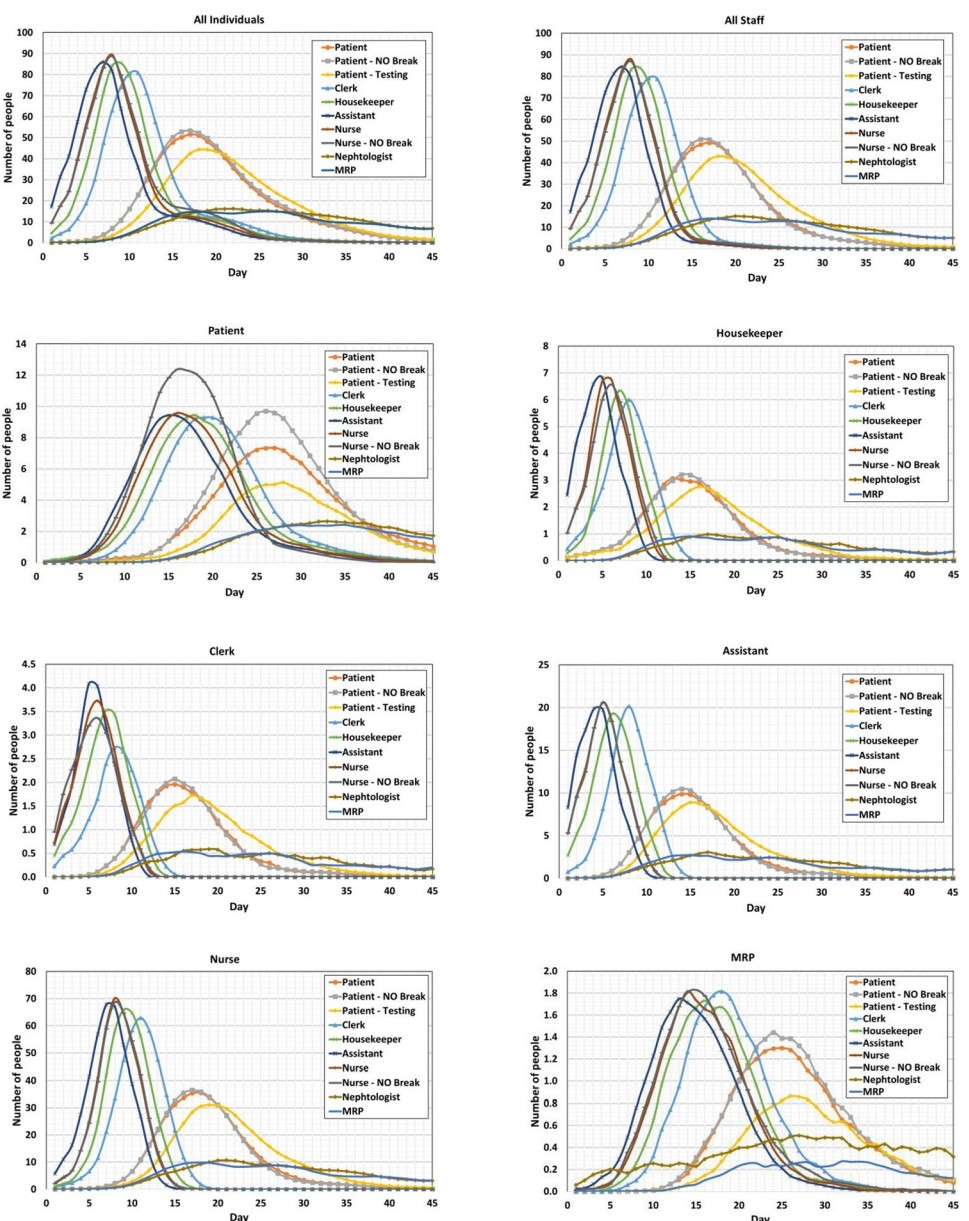

**Fig 8. Number of individuals exposed from a category (title of each chart) over time (day) under different scenarios (shown in the legend) such as: Entering an asymptomatic person, without break, and random testing in the dialysis unit.**

Another model was simulating disease transmission using the modified SEIR model that included asymptomatic, pre-symptomatic, and symptomatic components. The results show that under the base scenario that is with limited protection and in the early stages of the pandemic, an asymptomatic unchecked patient, nurse, assistant, and clerk can have a significant impact on disease propagation in the system due to the large social mix of agents involved in the dialysis process. This paper shows the capability of the agent-based simulation for modelling the spread of a disease through a dialysis unit and provides an opportunity to identify modifiable factors and behaviors that lead to increased infection rates. This would provide evidence-based targets to direct resources for virus transmission mitigation.

**Table 7. The maximum percent of individuals exposed and its corresponding time (day) when an asymptomatic patient or staff is coming to the dialysis unit.**

| Asymptomatic Agent > | | Pat. | Pat.NoBr. | Pat.Tes. | Cle. | Hou. | Assi. | Nur. | Nur.NoBr. | Nep. | MRP |
|---|---|---|---|---|---|---|---|---|---|---|---|
| All individual | Number | 11% | 11% | 9.1% | 16.7% | 18% | 18% | 18.3% | 18.3% | 3.3% | 3.1% |
| | Time (day) | 17 | 17 | 18 | 11 | 9 | 7 | 8 | 8 | 22 | 26 |
| All staff | Number | 28% | 28.7% | 24.2% | 44.4% | 47% | 47% | 49.4% | 48.9% | 8.4% | 7.9% |
| | Time (day) | 17 | 16 | 18 | 10 | 9 | 7 | 8 | 8 | 20 | 17 |
| Patients | Number | 2.3% | 3.2% | 1.6% | 2.9% | 2.9% | 2.9% | 3.2% | 3.9% | 1.0% | 0.6% |
| | Time (day) | 27 | 26 | 28 | 20 | 18 | 15 | 16 | 16 | 33 | 35 |
| Clerks | Number | 33% | 33% | 33% | 50% | 67% | 67% | 67% | 50.0% | 17% | 17% |
| | Time (day) | 15 | 15 | 17 | 8 | 7 | 5 | 6 | 6 | 19 | 16 |
| Housekeepers | Number | 30% | 30% | 30.0% | 60% | 60% | 70% | 70% | 70.0% | 10% | 10% |
| | Time (day) | 13 | 14 | 16 | 8 | 7 | 5 | 6 | 6 | 17 | 15 |
| Assistants | Number | 33% | 33% | 30.0% | 66.7% | 63% | 67% | 70% | 70.0% | 10% | 10% |
| | Time (day) | 14 | 14 | 15 | 8 | 6 | 4 | 5 | 5 | 17 | 15 |
| Nurses | Number | 33% | 33.6% | 28.2% | 57.3% | 60% | 62% | 64% | 62.7% | 10% | 9.1% |
| | Time (day) | 18 | 17 | 19 | 11 | 9 | 7 | 8 | 8 | 21 | 18 |
| Nephrologist | Number | 8% | 8% | 8% | 8% | 8% | 8% | 8% | 17% | 0.0% | 0.0% |
| | Time (day) | 24 | 24 | 26 | 16 | 15 | 14 | 15 | 16 | - | - |
| MRPs | Number | 10% | 10.0% | 10.0% | 20.0% | 20% | 20% | 20% | 20% | 10% | 0.0% |
| | Time (day) | 25 | 24 | 26 | 18 | 16 | 13 | 14 | 15 | 26 | - |

Our simulation covers a work flowchart of a typical dialysis unit in a normal condition and takes the distance-based direct contacts between individuals as the reason for disease transmission. Other parameters such as the density of virus in the air, ventilation system, and infected surfaces can affect the spread of disease transmission.

CDC defines close contact as: "*someone who was within 6 feet of an infected person for a cumulative total of 15 minutes or more over a 24-hour period starting from 2 days before the illness onset (or, for asymptomatic patients, 2 days prior to testing specimen collection) until the time the patient is isolated*" [37]. This is used as an operational definition for contact investigation due to the limited available data. Other parameters such as the type of symptoms of infected individuals and environmental factors (crowding, adequacy of ventilation, whether exposure was indoors or outdoors) can affect these values. In general, in a large population, there is no way or method to define the average number of individuals and infected individual meetings [38]. Sensitivity analysis and calibration of the mathematical models based on the real data are used to define these parameters for large populations. For example, Zeng, Guo [38] calculated the average number of contacts per exposed individual per day of 20 and 24 for these locations Hubei Province and Outside Hubei Province of China. Also, statistical analysis for closed environments can estimate the contact rate between the individuals. The contact rate among the people who are working or receiving treatments in a hospital varies depending on the type of work and hospitalization. Baek, Lee [4] used the value of contact duration of about 5 hours for the hospital wards and less than 0.5 hours for the outpatient department and emergency room for physicians, nurses, and other caregivers. Patients and caregivers in the hospital wards have the longest contact time. The contacts between the same category of agents have been assumed to be higher than the other probable contacts. Using the agent-based simulation, we calculated the time series of the number of contacts and time-length that people spend together in the dialysis unit (Table 6). The simulation showed that individuals who have the same station or are in the same category, have the maximum rate of contact and spend more time together. The contact parameters vary during a day and increase when the schedule

of staff is overlapped. Staff who are working in the same station are spending more than 15 minutes during a working shift and based on the CDC [37] definition, they will probably be exposed if one of them is infected.

It is important to highlight some of the limitations of the current study that we are currently working on. As our contacts matrices have been developed based on one dialysis unit with a specific workflow, more simulations of this type in other units need to be developed to examine if the same contacts patterns exist. Having said that our disease transmission model allows for new contacts matrices to be used. Moreover, the contacts matrices generated in this study need to be validated against some real measurements which require field observations and measurements that were not possible under the ongoing COVID-19 pandemic conditions.

## Conclusion

Results of the high-resolution calculation of contact parameters show a wide range of contact and various contact time among all agents during a workday which is important in virus transmission. Using this time series of contact we can minimize or optimize the workflow of the agents to have the lowest probability of the disease spread in hospitals. It is recommended that new studies on disease transmission consider these variations in contact parameters to find more realistic transmission parameters. The presented results here are based on some sample experiments which need to be expanded to include more variations of parameter values and virus protection measures. The current model does not capture all the complexity of interactions that are affecting disease transmission. During the pandemic, the UHN dialysis unit guidelines for preventing infection have changed several times based on emerging evidence and fortunately, the actual infection rates were low. The contact matrices generated by our simulation provide reasonable results when entered into the disease transmission model. We aim to continue further assessment and validate our model, to use it for policy and scenario analyses in the future. This study provides promising results for using agent-based simulation at a micro-scale to develop various contact matrices that can be used for detailed disease modelling and investigation of various mitigation measures. However, more studies like this and cross-validations are needed.

## Author Contributions

**Conceptualization:** Mohammadali Tofighi, Ali Asgary.

**Data curation:** Mohammadali Tofighi, Ali Asgary, Asad A. Merchant, Mohammad Ali Shafiee.

**Formal analysis:** Mohammadali Tofighi, Ali Asgary, Mahdi M. Najafabadi.

**Funding acquisition:** Ali Asgary.

**Investigation:** Mohammadali Tofighi, Ali Asgary, Asad A. Merchant, Mahdi M. Najafabadi.

**Methodology:** Mohammadali Tofighi, Ali Asgary.

**Project administration:** Ali Asgary.

**Resources:** Mohammadali Tofighi, Ali Asgary.

**Software:** Mohammadali Tofighi, Ali Asgary, Mahdi M. Najafabadi, Nazanin Nadri.

**Supervision:** Ali Asgary, Mohammad Ali Shafiee, Jane Heffernan, Jianhong Wu.

**Validation:** Mohammadali Tofighi, Ali Asgary.

**Visualization:** Mohammadali Tofighi, Ali Asgary.

**Writing – original draft:** Mohammadali Tofighi, Ali Asgary, Asad A. Merchant, Mahdi M. Najafabadi, Nazanin Nadri, Mehdi Aarabi.

**Writing – review & editing:** Mohammadali Tofighi, Ali Asgary, Asad A. Merchant, Mohammad Ali Shafiee, Mahdi M. Najafabadi, Jane Heffernan, Jianhong Wu.

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
