## [Decision Letter · Decision Letter 0]

10 Sep 2021

PONE-D-21-07385Modelling COVID -19 transmission in a hemodialysis centre using simulation generated contacts matricesPLOS ONE

Dear Dr. Tofighi,

Thank you for submitting your manuscript to PLOS ONE. After careful consideration, we feel that it has merit but does not fully meet PLOS ONE’s publication criteria as it currently stands. Therefore, we invite you to submit a revised version of the manuscript that addresses the points raised during the review process.

We look forward to receiving your revised manuscript.

Kind regards,

Michele Provenzano

Academic Editor

PLOS ONE

Journal Requirements:

Additional Editor Comments (if provided):

Please carefully read the Reviewers' comments and try to answer all the criticisms

Reviewers' comments:

Reviewer's Responses to Questions

**Comments to the Author**

1. Is the manuscript technically sound, and do the data support the conclusions?

Reviewer #1: Yes

Reviewer #2: Yes

2. Has the statistical analysis been performed appropriately and rigorously? 

Reviewer #1: Yes

Reviewer #2: Yes

3. Have the authors made all data underlying the findings in their manuscript fully available?

Reviewer #1: Yes

Reviewer #2: Yes

4. Is the manuscript presented in an intelligible fashion and written in standard English?

Reviewer #1: Yes

Reviewer #2: Yes

5. Review Comments to the Author

Reviewer #1: This very interesting paper provides relevant results regarding the operations of a typical large dialysis unit and generate contact matrices to examine outbreak scenarios

during COVID-19 pandemic. The authors presented the details of the contact matrix generation process and demonstrate how the simulation calculates a micro-scale contact matrix comprising the number and duration of

contacts at a micro-scale time step. They used the contacts matrix in an agent-based model to predict disease transmission under different scenarios. The results show that micro-simulation can be used to estimate contact matrices, which can be used effectively for disease modelling in dialysis and similar settings. The manuscript is well written and organized. Methods are appropriate, results are clearly described and illustrated, as well as properly discussed. References are relevant and updated; however it could be useful to discuss the impact of cardiovascular risk factors in chronic kidney disease during COVID-19 pandemic, referring to the following articles doi:10.31083/j.rcm.2019.04.548, https://doi.org/10.1159/000514467. Therefore, this paper requires minor corrections and can be very useful for Plos One readers, because it provides very interesting information within the current context of published studies.

Reviewer #2: I think the article is of interest and reflects the classic canons. demonstrates the goal set by the study. I will hence suggest the manuscript in this form. It respects research and pubilcation ethics.

6. PLOS authors have the option to publish the peer review history of their article (what does this mean?). If published, this will include your full peer review and any attached files.

Reviewer #1: No

Reviewer #2: **Yes: **Provenzano D

---

## [Author Response · Author response to Decision Letter 0]

14 Oct 2021

Response to the Reviewers’ Comments

on the Manuscript Entitled “Modelling COVID -19 transmission in a hemodialysis centre using simulation generated contacts matrices”

Submitted for publication in PLOS ONE [PONE-D-21-07385]

Mar 5, 2021

We are grateful to the Editor and to the Reviewers for carefully reviewing our paper, which has been revised according to your helpful comments. We believe that your suggestions have enhanced the quality of our paper. Our responses and revisions are given below, with the editor’s and reviewers’ original comments in italics.

Review Comments to the Author

Reviewer #1: This very interesting paper provides relevant results regarding the operations of a typical large dialysis unit and generate contact matrices to examine outbreak scenarios

during COVID-19 pandemic. The authors presented the details of the contact matrix generation process and demonstrate how the simulation calculates a micro-scale contact matrix comprising the number and duration of contacts at a micro-scale time step. They used the contacts matrix in an agent-based model to predict disease transmission under different scenarios. The results show that micro-simulation can be used to estimate contact matrices, which can be used effectively for disease modelling in dialysis and similar settings. The manuscript is well written and organized. Methods are appropriate, results are clearly described and illustrated, as well as properly discussed. References are relevant and updated; however it could be useful to discuss the impact of cardiovascular risk factors in chronic kidney disease during COVID-19 pandemic, referring to the following articles doi:10.31083/j.rcm.2019.04.548, https://doi.org/10.1159/000514467. Therefore, this paper requires minor corrections and can be very useful for Plos One readers, because it provides very interesting information within the current context of published studies.

Response:

Thank you for these affirmative and constructive comments. The mentioned impact has been highlighted and referenced in lines 82 - 84 of the revised version as follows:

“Furthermore, chronic kidney disease (CKD) can be considered as a cardiovascular risk equivalent, and COVID-19 that enhances the risk of cardiovascular events may synergize with pre-existing cardiovascular risk factors in CKD patients [6, 7].”

Reviewer #2: I think the article is of interest and reflects the classic canons. demonstrates the goal set by the study. I will hence suggest the manuscript in this form. It respects research and publication ethics.

Response:

Thank you for the affirmative and supportive comment.

---

## [Editor Report · Decision Letter 1]

2 Nov 2021

Modelling COVID -19 transmission in a hemodialysis centre using simulation generated contacts matrices

PONE-D-21-07385R1

Dear Dr. Tofighi,

We’re pleased to inform you that your manuscript has been judged scientifically suitable for publication and will be formally accepted for publication once it meets all outstanding technical requirements.

Kind regards,

Michele Provenzano

Academic Editor

PLOS ONE
---

## [Editor Report · Acceptance letter]

10 Nov 2021

PONE-D-21-07385R1 

Modelling COVID -19 transmission in a hemodialysis centre using simulation generated contacts matrices 

Dear Dr. Tofighi:

I'm pleased to inform you that your manuscript has been deemed suitable for publication in PLOS ONE. Congratulations! Your manuscript is now with our production department. 

Kind regards, 

on behalf of

Dr. Michele Provenzano 

Academic Editor

PLOS ONE